

# Enhancing rice yield prediction: a deep fusion model integrating ResNet50-LSTM with multi source data

Aqsa Aslam and Saima Farhan

Department of Computer Science, Lahore College for Women University, Lahore, Punjab, Pakistan

## ABSTRACT

Rice production is pivotal for ensuring global food security. In Pakistan, rice is not only the dominant Kharif crop but also a significant export commodity that significantly impacts the state's economy. However, Pakistan faces challenges such as abrupt climate change and the COVID-19 pandemic, which affect rice production and underscore the need for predictive models for informed decisions aimed at improving productivity and ultimately the state's economy. This article presents an innovative deep learning-based hybrid predictive model, ResNet50-LSTM, designed to forecast rice yields in the Gujranwala district, Pakistan, utilizing multi-modal data. The model incorporates MODIS satellite imagery capturing EVI, LAI, and FPAR indices along with meteorological and soil data. Google Earth Engine is used for the collection and preprocessing of satellite imagery, where the preprocessing steps involve data filtering, applying region geometry, interpolation, and aggregation. These preprocessing steps were applied manually on meteorological and soil data. Following feature extraction from the imagery data using ResNet50, the three LSTM model configurations are presented with distinct layer architectures. The findings of this study exhibit that the model configuration featuring two LSTM layers with interconnected cells outperforms other proposed configurations in terms of prediction performance. Analysis of various feature combinations reveals that the selected feature set (EVI, FPAR, climate, and soil variables) yields highly accurate results with an $R^2 = 0.9903$, RMSE = 0.1854, MAPE = 0.62%, MAE = 0.1384, MRE = 0.0062, and Willmott's index of agreement = 0.9536. Moreover, the combination of EVI and FPAR is identified as particularly effective. Our findings revealed the potential of our framework for globally estimating crop yields through the utilization of publicly available multi-source data.

# INTRODUCTION

Rice production holds significant global importance, with Asia heavily reliant on it as a dietary staple. In crop year 2022–2023, global rice consumption reached 517,184 thousand metric tons, prompting increased rice imports in many countries by which rice-exporting nations experience economic benefits through international trade (*Statista, 2024*). Pakistan, possessing with tremendous agricultural potential owing to its fertile plains,

Corresponding author
Saima Farhan,
saima.farhan@lcwu.edu.pk

diverse agroclimatic zones, and abundant water resources, has a robust agricultural sector, divided into Rabi and Kharif cropping seasons-based on the monsoon rainfall pattern and temperature fluctuations. Rice, a Kharif crop, is prominently cultivated in Punjab and Sindh provinces. Pakistan produced 9,322.67 million tons of rice in 2022, with Punjab contributing 5,779 million tons, highlighting its significant role in rice production (*CRS, 2022*). Globally, rice is a highly traded commodity, with Pakistan ranking fourth in rice exports. In 2021, Pakistan exported around 3.9 million tons of rice (*FAOSTAT, 2024*). Rice cultivation significantly impacts Pakistan's economy, contributing to foreign exchange revenue, employment, gross domestic product (GDP), government revenue through taxes on rice exports. Challenges such as labor shortages and disruptions in the supply chain due to COVID-19 pandemic (*Goeb et al., 2022*), locust attacks (*Usman et al., 2022*), losses from flooding and climate change affect farmers and their crops (*Shakoor et al., 2015*). The World Bank anticipates up to nine million people in Pakistan may fall into poverty due to the pandemic (*Saifi & Horowitz, 2023*). To address these challenges, Pakistan needs to focus on sustainable rice production. Predicting rice yields can assist farmers in making informed decisions regarding inputs and planning, thereby enhancing productivity and reducing costs on resource allocation and management practices.

Historically, rice yield prediction methods have been limited by factors such as reliance on field surveys and visual inspections, leading to inconsistent and sometimes inaccurate results due to human errors and variability among surveyors. Alternative approaches include physiological models or crop growth models which simulate rice plant development based on environmental variables, *e.g.*, phenology-based models and process-based models (*Sivakumar et al., 2003*), and statistical approaches which analyze past data to identify trends and patterns, assuming that historical trends will persist into the future. However, if underlying assumptions are not satisfied or if the data is not a good reflection of the current situation, these models may struggle with accuracy and scalability. Assimilation modeling combines actual data with model simulations to improve prediction accuracy, but assumptions and computational complexities make scaling difficult.

An alternative approach, machine learning (ML), presents a strategy to learn patterns and relationships directly from data without the need for explicit models or presumptions. ML is capable of processing unstructured data and capturing complex relationships that conventional methods may find challenging to comprehend. However, ML often requires human effort for feature engineering and can introduce bias if not handled carefully. Deep learning (DL), a subset of ML (*Lecun, Bengio & Hinton, 2015*), automates the process of feature extraction from raw data, reduces biasness and provides highly accurate end-to-end predictions.

Remote sensing, using aircraft, satellites, unmanned aerial vehicles (UAVs) equipped with specialized sensors (*Campbell & James, 2011*), and alternate platforms, emerges as another valuable tool for acquisition of comprehensive data on environmental conditions and crop health. This technique aids in ecological research by enabling the tracking and examining of Earth's physical properties. Satellite imagery, provided by satellite-based remote sensing tools such as Sentinel-1, Sentinel-2, Landsat, and MODIS (Moderate

Resolution Imaging Spectroradiometer), offers detailed insights into land cover, weather patterns, and terrain. Google Earth Engine (GEE), a cloud-based geospatial analysis tool, facilitates the processing and analysis of such satellite data, improving remote sensing capabilities for monitoring agricultural landscapes and predicting rice yields.

A surge of interest has been observed in recent agricultural research targeted at enhancing crop yield. Notable subjects include crop yield prediction (*Muruganantham et al., 1990*; *Oikonomidis, Catal & Kassahun, 2023*), crop loss estimation (*Wang et al., 2022*), pest and disease management (*Gokila & Santhi, 2022*), soil health and fertility (*Swapna, Manivannan & Kamalahasan, 2022*), *etc*. Several studies have targeted the prediction of specific crop such as wheat (*Fei et al., 2023*), cotton (*Tahseen Haider et al., 2022*), maize (*Zhang et al., 2019*), and others.

For rice yield, some studies have exclusively trained models on meteorological data and yield statistics (*Chu & Yu, 2020*), whereas others incorporated sentinel imagery (*Fernandez-Beltran et al., 2021*), or Landsat imagery data (*Siyal, Dempewolf & Becker-Reshef, 2015*). Some explored MODIS satellite variables including normalized difference vegetation index (NDVI) (*Son et al., 2020*), enhanced vegetation index (EVI) (*Cao et al., 2021*; *Zhou, Xu & Chen, 2023*), solar-induced chlorophyll fluorescence (SIF) (*Cao et al., 2021*; *Liu et al., 2022*), gross primary productivity (GPP), soil-adjusted vegetation index (SAVI) (*Zhou, Xu & Chen, 2023*), near-infrared reflectance of vegetation (NIRv) (*Liu et al., 2022*), *etc*. However, numerous satellite variables remain untapped, and there is potential to integrate extensive abiotic information with the high spatial and temporal coverage of satellite images.

In rice yield prediction, ML approaches have been utilized, such as bagging of decision trees (*Siyal, Dempewolf & Becker-Reshef, 2015*), random forest (RF), and support vector machine (SVM) (*Son et al., 2020*). A study (*Chu & Yu, 2020*) fuses two neural networks *i.e.*, back-propagation neural networks (BPNNs) to learn deep spatial features from area data, and independently recurrent neural network (IndRNN) to learn temporal features from meteorological data. An informer transformer-based model has also been employed (*Liu et al., 2022*). The use of a 3D convolutional neural network (3DCNN) on sentinel images (*Fernandez-Beltran et al., 2021*), and the presentation of deep learning based long short-term memory (LSTM) model (*Cao et al., 2021*; *Zhou, Xu & Chen, 2023*) demonstrate the diversity of approaches. But still the opportunity exists for improvement in DL models, given their early development.

This study aims to predict rice yield across four tehsils of Gujranwala district, Pakistan, using satellite imagery, meteorological data, and soil data collected from 2000 to 2021. The data undergoes comprehensive preprocessing and alignment. The study utilizes two models: residual network 50 (ResNet50), which is employed to extract essential features from satellite images, and three distinct configurations of LSTM models are proposed. The objective of this study is to address several questions: (i) which LSTM model configuration among the three proposed provides accurate and precise results? (ii) which combination of features affects the predictions? (iii) which satellite variable has the most significant impact on predictions? and (iv) to investigate the convergence ability of the model.

## MATERIALS AND METHODS

This section is organized into five subsections: (i) study area, (ii) description of datasets acquired from various sources, (iii) preprocessing techniques applied to the datasets, (iv) the proposed deep learning model, and (v) evaluation metrics used in this study to validate the proposed model.

### Study area

The region of interest of this study is the rice cultivation area of Gujranwala district, Punjab, Pakistan. Gujranwala is situated between 31.8°N–32.5°N and 73.6°E–74.5°E, experiencing a hot climate during the Kharif season, with temperature ranging from 30 °C to 42 °C, a UV index from 1.5 to 3, precipitation from 2 to 20 mm per day, humidity ranging from 45% to 80%, and wind speeds from 1 to 3 m per second. Additionally, the soil in this region has an average pH level of approximately 8.42, which supports crop cultivation. Rice is the predominant Kharif crop in Gujranwala district, with the growth period extended from late-April to mid-November. According to the government of Punjab, it was reported that rice was cultivated in this region on 643 million acres in 2021 (*Crops Statistics, 2022*). Gujranwala is recognized as the bread basket of Pakistan. This study encompasses the four tehsils (counties) of Gujranwala district: Wazirabad, Gujranwala city, Nowshera Virkan, and Kamoke. Figure 1 illustrates the location and spatial distribution of the region of interest.

### Dataset description

In this study, we gathered yield statistics, remotely sensed satellite data, environmental data, and some auxiliary data from various sources. An overview of the collected data is presented in Table 1. All data were collected for the period 2000–2021 from last-April to mid-November.

#### *Yield statistics*

Average annual yield data in terms of maunds per acre for Gujranwala district were obtained for the period 2000–2021 from the crop reporting service, Government of Punjab, Pakistan (*CRS, 2022*). This information was further validated using agriculture statistics from the Pakistan Bureau of Statistics. In this study, yield data were served as annotations for training and validating the proposed model.

#### *Satellite imagery*

This study utilized three NASA Terra-MODIS products, which were extracted from the GEE. A collaborative effort between NASA and GEE ensures the ingestion of satellite images on a daily basis, making them globally accessible for data mining.

One of the MODIS products used in this study is EVI which is an enhanced version of NDVI. EVI is designed to quantify vegetation greenness. NDVI is considered to be a traditional vegetation index, whereas EVI has improved sensitivity to biomass regions and the ability to reduce atmospheric influences. EVI is formulated as given in Eq. (1).

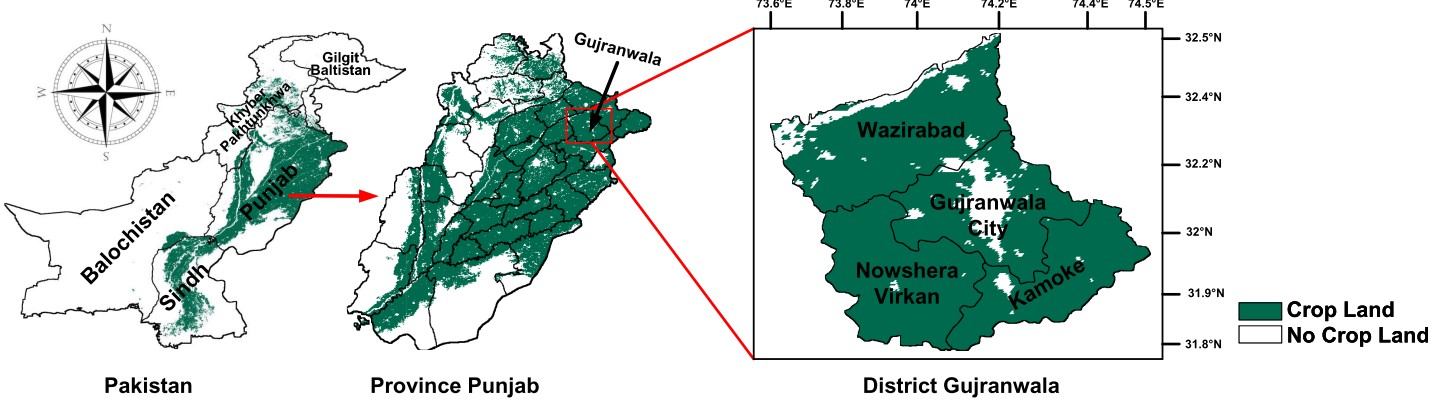

**Figure 1 Location and spatial distribution of crop producing areas of Gujranwala district, Pakistan.** Map data © 2023 Google Earth Engine. Dataset source: Pakistan-Subnational Administrative Boundaries; The Humanitarian Data Exchange; CC BY 4.0, https://creativecommons.org/licenses/by/4.0.

$$EVI = 2.5 \times \frac{NIR - R}{NIR + C_1(R) - C_2(B) + L} \qquad (1)$$

where NIR is near-infrared reflectance, R is red reflectance, B is blue reflectance, $C_1$ and $C_2$ are coefficients, usually set to 6 and 7.5 respectively, and L is the canopy background adjustment, generally set to 1. We extracted the EVI band from MOD13A2H.061 collection using GEE, which provides pre-calculated EVI data in raster format compositing at 16-day intervals with the spatial resolution of 1 km (*Didan, 2021*).

The other two products, leaf area index (LAI) and fraction of photosynthetically active radiation (FPAR), were also extracted. LAI represents the amount of leaf material per unit ground area and is closely associated with the plant's evapotranspiration capacity. Whereas, FPAR represents the fraction of photosynthetically active radiations absorbed by green vegetation. They are expressed in Eqs. (2) and (3) respectively.

$$LAI = \frac{1}{c} \times \ln\left(\frac{PAR_{canopy}}{PAR_{sun}} \times (1 - R_g)\right) \qquad (2)$$

$$FPAR = 1 - e^{-0.5(LAI)} \qquad (3)$$

where c is the extinction coefficient which represents the fraction of radiation absorbed per unit LAI, $PAR_{canopy}$ is the photosynthetically active radiation (PAR) within the canopy, $PAR_{sun}$ is the incoming PAR, and $R_g$ is the fraction of incoming radiation absorbed by the ground. These data were extracted from MOD15A2H.061 collection, which provides 8-day composite data with the spatial resolution of 500 m (*Myneni, Knyazikhin & Park, 2021*).

### Climate and soil data

The climate and soil parameters utilized in this study is sourced from the NASA Langley Research Center (LaRC) Prediction of Worldwide Energy Resources (POWER) project funded through the NASA Earth Science/Applied Science Program. This project was initiated to enhance earth science datasets and to generate new datasets from emerging

**Table 1 Description of dataset collected from multiple sources.**

| Category | Attributes (Resolution) | Time steps | Source |
|---|---|---|---|
| Yield statistics | Average yield (maund/acre) | Annual | Crop reporting service, Government of the Punjab, Pakistan (crs.agripunjab.gov.pk) |
| Satellite images | EVI (1,748 × 1,317) | 16 days | MODIS dataset MOD13A2.061 (GEE) |
| | FPAR (1,748 × 1,317) | 8 days | MODIS dataset MOD15A2H.061 (GEE) |
| | LAI (1,748 × 1,317) | 8 days | MODIS dataset MOD15A2H.061 (GEE) |
| Climate data | Clear sky surface PAR total (W/m$^2$) | Daily | power.larc.nasa.gov |
| | All sky surface UV index (dimensionless) | | |
| | Temperature at 2 m (C) | | |
| | Dew/Frost point at 2 m (C) | | |
| | Wet bulb temperature at 2 m (C) | | |
| | Earth skin temperature (C) | | |
| | Temperature at 2 m Maximum (C) | | |
| | Temperature at 2 m Minimum (C) | | |
| | Specific humidity at 2 m (g/kg) | | |
| | Relative humidity at 2 m (%) | | |
| | Precipitation corrected (mm/day) | | |
| | Surface pressure (kPa) | | |
| | Wind speed at 2 m (m/s) | | |
| | Wind direction at 2 m (degrees) | | |
| Soil data | Surface soil wetness | Daily | Power.larc.nasa.gov |
| | Root zone soil wetness | | |
| | Profile soil moisture | | |
| Auxiliary data | Crop mask | Annual | MODIS land cover type MCD12Q1.006 (GEE) |
| | Shape file of Pakistan (Tehsil-wise) | – | Pakistan-subnational administrative boundaries (https://data.humdata.org/dataset/cod-ab-pak?) |

satellite systems (*POWER, 2020*). The POWER project offers a data access viewer interface. Through this interface, data specifically tailored for the agroclimatology community was extracted, focusing on the daily temporal average within our study area.

From the POWER project, climate parameters were extracted, encompassing clear sky surface PAR total, all-sky surface ultraviolet (UV) index, temperature (minimum, maximum, wet bulb, earth skin) at 2 m, dew/frost point at 2 m, humidity (relative, specific) at 2 m, precipitation, surface pressure, wind speed, and wind direction at 2 m. Additionally, soil parameters retrieved from the POWER project include surface soil wetness, root zone soil wetness, and profile soil moisture.

### Auxiliary data

Certain auxiliary and supporting data utilized in this study include a MODIS land cover type product, MCD12Q1.006, employed for crop land masking. This product was extracted from GEE at a spatial resolution of 500 m and is presented in the yearly composites (*Friedl & Sulla-Menashe, 2019*). Furthermore, the shapefile containing the

boundaries of all tehsils in the Gujranwala district was acquired from the Pakistan subnational administrative boundaries dataset sourced by 'The Humanitarian Data Exchange' (*Humanitarian Data Exchange, 2022*).

## Data preprocessing

This study integrates two data modalities-satellite data in the form of images and meteorological and soil data in numeric format–the preprocessing approach differs for each. All imagery data underwent preprocessing using Java scripting in the GEE platform, whereas numerical data were preprocessed manually.

Initially, all data was filtered for the time period spanning from 2000 to 2021, especially for the months from April to November. Subsequently, the region geometry was applied to the image data to encompass all tehsils of the Gujranwala district, as obtained from the Pakistan subnational administrative boundaries. Nearest-neighbor interpolation was also employed to address missing data within the LAI and FPAR collections. The EVI collection was already complete, eliminating the need for interpolation. However, some samples of meteorological and soil data were also missing; therefore, interpolation was performed to identify missing values. The interpolation is formulated as given in Eq. (4).

$$Z = \begin{cases} f(x_i, y_i) \approx f(x_{i-1}, \ y_{i-1}) & \textit{for imagery data} \\ f(x_i) \approx f(x_{i-1}) & \textit{for numerical data} \end{cases} \tag{4}$$

where i represents missing sample and i-1 represents the previous sample of the missing sample.

Moreover, meteorological and soil data were acquired on a daily basis, while LAI and FPAR are composites of 8 days, and EVI is a composite at a 16-day interval. In this study, all datasets were aligned at a 16-day interval, resulting in 13 time-steps. The start and end dates of 13 time-steps used in this study for leap and non-leap years are illustrated in Table 2.

A crop mask (MCD12Q1) was applied to all satellite images-EVI, LAI, FPAR images-extracting only the crop cultivated areas within the region. Before applying the mask, they went through interpolation. Despite LAI and FPAR inherently comprising cultivated areas only, an additional crop mask was applied to enhance precision. It is noteworthy that approximately 84% of the cultivated land in Gujranwala is dedicated to rice cultivation, making rice the dominant crop in the Kharif season. Given this dominance, rice crop masking was not applied, as it would not have a significant impact. The applied preprocessing steps of this study are outlined in Fig. 2.

## Model architecture

This study proposes a hybrid of two deep learning models: (i) ResNet50 for feature extraction and feature selection of satellite imagery, and (ii) three distinct architectures of LSTM for predicting rice yields.

### ResNet50 architecture

The ResNet50 architecture is a transfer learning based convolutional neural network (CNN) comprising 50 layers. It includes initial layers, residual blocks, and final layers. The

**Table 2 Time-steps at 16-day interval used in this study.**

| Intervals | For non-leap years | For leap years |
|---|---|---|
| 1 | 24 April–9 May | 23 April–8 May |
| 2 | 10 May–25 May | 9 May–24 May |
| 3 | 26 May–10 June | 25 May–9 June |
| 4 | 11 June–26 June | 10 June–25 June |
| 5 | 27 June–12 July | 26 June–11 July |
| 6 | 13 July–28 July | 12 July–27 July |
| 7 | 29 July–13 August | 28 July–12 August |
| 8 | 14 August–29 August | 13 August–28 August |
| 9 | 30 August–14 September | 29 August–13 September |
| 10 | 15 September–30 September | 14 September–29 September |
| 11 | 1 October–16 October | 30 September–15 October |
| 12 | 17 October–1 November | 16 October–31 October |
| 13 | 2 November–17 November | 1 November–16 November |

residual blocks, which consist of convolutional and identity blocks, allow the network to learn significantly deeper patterns without encountering the vanishing gradient problem. This problem arises when the gradient of the loss function becomes very small relative to the network parameters, which propagates backward through the network during training, severely slowing down the training process and making effective training difficult. ResNet50 mitigates this problem by employing skip connections in the residual blocks, allowing information to bypass one or more layers, thereby introducing the concept of residual learning. This pre-trained model has been trained on the ImageNet dataset (*Koonce, 2021*).

In this study, the weights from ImageNet were utilized for feature extraction. Following the extraction of features, this model was employed to select 1,000 most prominent features for each image by tailoring the last layer of the ResNet50 model. These selected features of satellite images are fed to the LSTM model. The architecture of ResNet50 model utilized in this study (*Mukherjee, 2022*) is depicted in Fig. 3.

### LSTM architecture

LSTM model is a type of recurrent neural network (RNN) designed to handle both spatial and temporal details. Unlike conventional RNN models that struggle with the vanishing gradient problem, LSTMs use gate functions to effectively learn and process long dependencies in sequential data (*Yu et al., 2019*). A LSTM cell comprises several key components: the cell state which serves as long-term memory, and three gates–the input gate, the forget gate, and the output gate. The input gate controls which new information should be updated or added to the cell state, the forget gate controls which information from the previous cell state should be removed or discarded. And the output gate controls the output based on the current cell state. The cell state is updated by combining the retained information from the previous state with new values, modulated by the input gate,

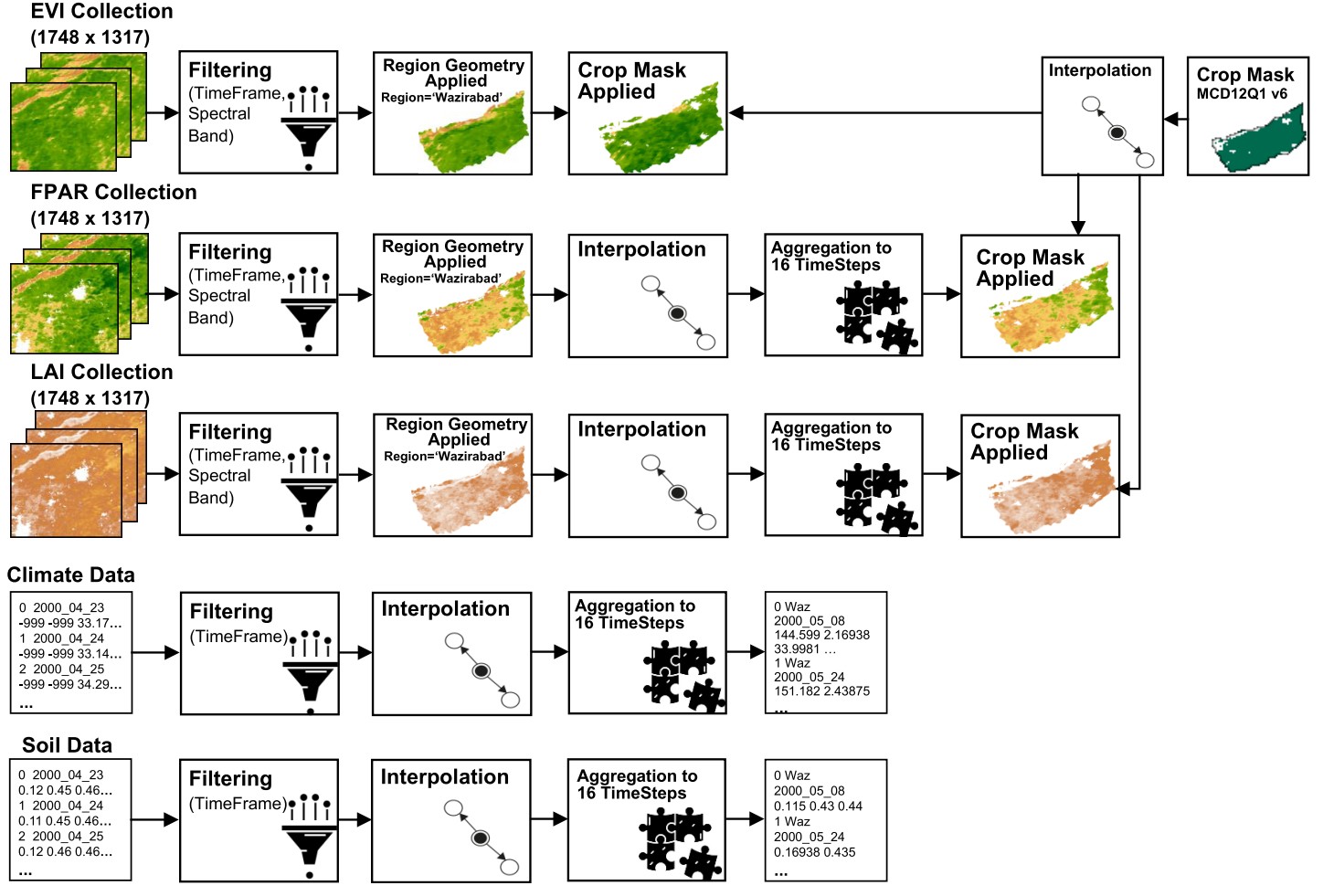

**Figure 2 Pre-processing flow on datasets.** Satellite imagery, crop mask © 2023 Google Earth Engine; Dataset source: MOD13A2, MOD15A2H; NASA LP DAAC at the USGS EROS center; Region geometry © 2023 Google Earth Engine; Dataset source: Pakistan-Subnational Administrative Boundaries; The Humanitarian Data Exchange; CC BY 4.0, https://creativecommons.org/licenses/by/4.0; Climate and soil data © *POWER, 2020*.

and by discarding unnecessary information as directed by the forget gate. The insights into a LSTM cell are depicted in Fig. 4.

### *Proposed configurations of the ResNet50-LSTM*

The satellite images undergo initial processing through the ResNet50 model, which extracts features from them using pre-trained weights of ImageNet. Subsequently, 1,000 most contributing features against each image were selected. These selected features are then input into proposed LSTM model for rice yield prediction. Conversely, numerical environmental data is directly fed into the LSTM model without undergoing the ResNet50 model.

This article proposes three configurations of the LSTM architecture for rice yield prediction, as depicted in Fig. 5. In all configurations, the features of satellite images–EVI, LAI, FPAR images–extracted from the ResNet50 model, and environmental data–climate

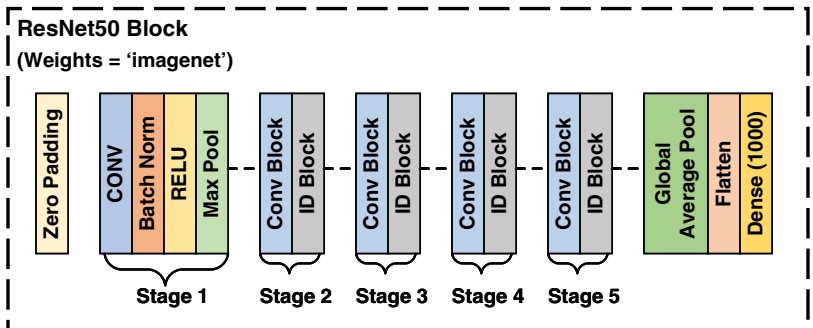

**Figure 3 Architecture of ResNet50.**

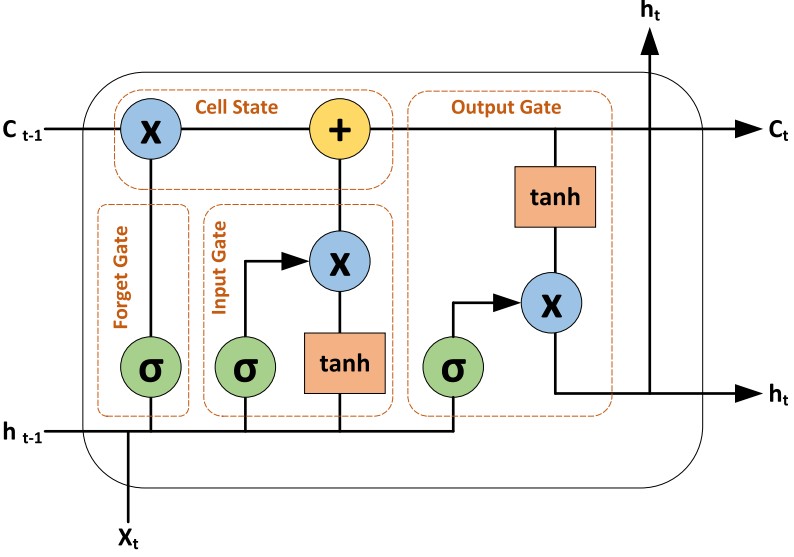

**Figure 4 Insights into the LSTM cell.**

and soil data–are input to five separate cells of LSTM at layer 1. Each LSTM cell encompasses various internal components, such as input, forget, and output gates, and a memory cell or cell state. The capacity and complexity of LSTM cell are measured in terms of units. In proposed architectures, all the LSTM cells are configured with 64 units, except for the cell that receives soil data, which has 32 units. Because soil data has less dimensionality as compared to other data.

In the first configuration shown in Fig. 5A, all the LSTM cells are independent. The results of these cells are then concatenated, followed by a dropout layer with a 20% dropout of neurons to reduce overfitting. Finally, a dense layer is set up to predict rice yield. In the second configuration shown in Fig. 5B, all five cells of LSTM at layer 1 are dependent on the result of their previous cell. For instance, LSTM cell 1 receives the EVI features, LSTM cell 2 receives the FPAR features along with the result of LSTM cell 1, and so on. The rest of the architecture remains consistent with Fig. 5A. In the third proposed configuration illustrated in Fig. 5C, all LSTM cells at layer 1 are interconnected and dependent on the

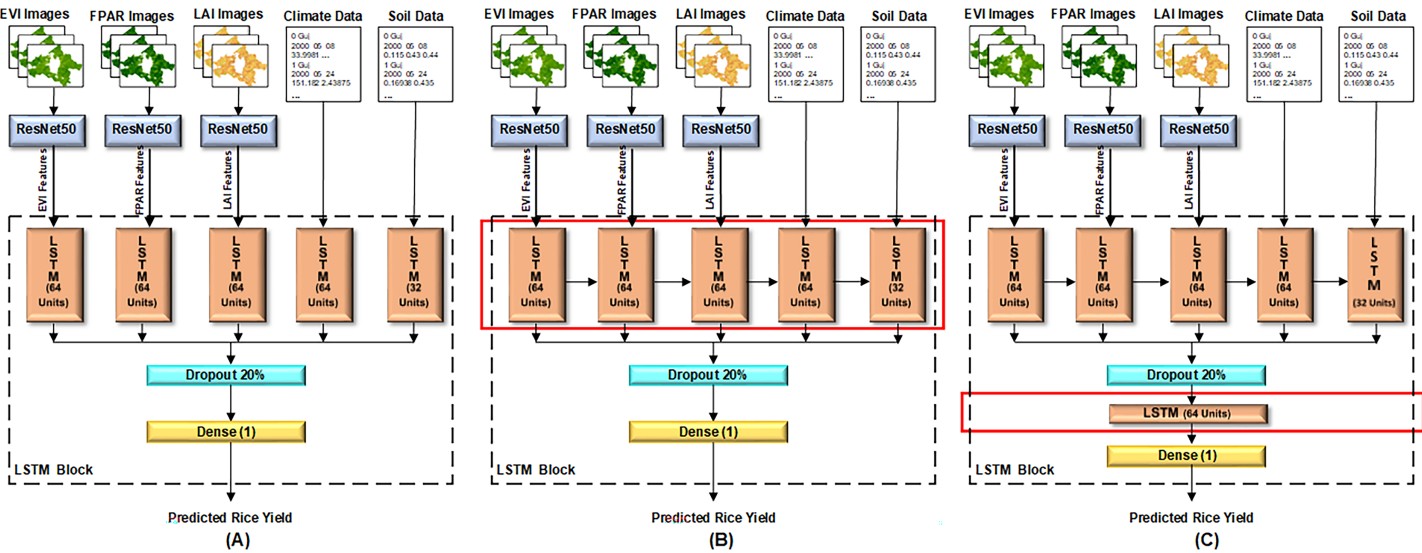

**Figure 5** **Three configurations of proposed LSTM model.** (A) Model configuration 1: single LSTM layer without interconnected cells, (B) model configuration 2: single LSTM layer with interconnected cells, (C) model configuration 3: dual LSTM layers with interconnected cells.

result of their previous cell. Additionally, another LSTM layer is added after the dropout layer, while the rest of the architecture remains unchanged. The changes in the layer architecture of the three configurations, as elaborated above, are highlighted with red boxes in Fig. 5.

The LSTM and ResNet50 architectures are developed within the Spyder notebook environment using the Keras (*Ketkar, 2017*), Numpy (*McKinney, 2013*), TensorFlow (*Singh & Manure, 2020*), and Sklearn (*McKinney, 2013*) libraries. The Adam optimizer is used to facilitate an adaptive learning rate. The proposed model is evaluated over ten-fold cross validation, and 100 epochs with a batch size set to 32.

## Evaluation methods

Obtaining real-world future data for validating the proposed model is not possible. Therefore, the evaluation of the proposed model is conducted using a k-fold cross validation technique. Initially, the dataset and yield values are shuffled as a group. The entire dataset is then divided into k equal-sized subsets referred to as folds. The model is trained and validated k times, utilizing a different fold as the validation set at each iteration (*Anguita et al., 2012*). In this study, k is set to 10. Various evaluation metrics are computed at each fold to assess the model's performance. Subsequently, the average of each evaluation metric across all folds is calculated to determine the generalized performance of the model.

Given that rice yield prediction involves predicting numerical outcomes, it is inherently a regression problem. Thus, the evaluation metrics utilized in this study include the coefficient of determination ($R^2$), root mean square error (RMSE), mean absolute error

(MAE), mean bias error (MBE), mean relative error (MRE), mean absolute percentage error (MAPE), and Willmott's index of agreement (d). These metrics are specifically designed to evaluate the performance of regression models, and are particularly chosen for this study as they collectively offer a well-rounded evaluation of different aspects of model performance, providing a comprehensive assessment of the model's predictive accuracy and reliability. These metrics are formulated in equations from Eqs. (5) to (11), respectively.

$$R^2 = 1 - \frac{\sum_{i=1}^{n} (R_i - P_i)^2}{\sum_{i=1}^{n} (R_i - mean_R)^2} \tag{5}$$

$$RMSE = \sqrt{\frac{\sum_{i=1}^{n} (R_i - P_i)^2}{n}} \tag{6}$$

$$MAE = \frac{\sum_{i=1}^{n} |R_i - P_i|}{n} \tag{7}$$

$$MBE = \frac{\sum_{i=1}^{n} (R_i - P_i)}{n} \tag{8}$$

$$MRE = \frac{1}{n} \sum_{i=1}^{n} \left| \frac{R_i - P_i}{R_i} \right| \tag{9}$$

$$MAPE = \frac{1}{n} \sum_{i=1}^{n} \left| \frac{R_i - P_i}{R_i} \right| \times 100 \tag{10}$$

$$d = 1 - \frac{\sum_{i=1}^{n} (R_i - P_i)^2}{\sum_{i=1}^{n} (|R_i - mean_R| + |P_i - mean_R|)^2} \tag{11}$$

where n denotes the total number of samples, $R_i$ represents the reported or observed value, $P_i$ represents the predicted value determined by the model, and $mean_R$ indicates the mean of all reported values. $R^2$ determines the proportion of variance in predicted values. It measures how well the data fit the regression model. The $R^2$ values vary from 0 to 1, with one indicating the perfect model fit. RMSE is the square root of the average of squared differences between reported and predicted values. It provides a clear measure of the model accuracy and is sensitive to large errors. MAE characterizes the average absolute differences between reported and predicted values, and provides a straightforward measure of prediction accuracy. Smaller values of RMSE, and MAE depicts the better model performance. MBE indicates whether the model tends to underpredict or overpredict. A positive MBE suggests overprediction, whereas a negative MBE suggests underprediction. MRE represents the average of the absolute values of the differences between predicted and reported values, and is useful for understanding the average relative size of the errors. MAPE provides a percentage-based measure of the average prediction error. Willmott's d measures the agreement between reported and predicted values, considering both the mean square error and variance of the reported values. It is a comprehensive measure that combines aspects of correlation and error magnitude. d ranges from 0 to 1, where one indicates perfect agreement between reported and predicted values, and suggest better model performance.

# RESULTS

This section is organized into two subsections: the description of feature sets and their integration, and the evaluation of proposed LSTM architecture across the selected feature sets.

## Feature sets and their integration

The LSTM accepts two modalities of data: imagery data and numerical data. The features from imagery data are extracted from proposed tailored ResNet50 model. One complete input sample comprises of five sets of input features: (i) a set of 1,000 features extracted from EVI images, (ii) a set of 1,000 features extracted from LAI images, (iii) a set of 1,000 features extracted from FPAR images, (iv) a set of 14 climate variables, and (v) a set of 3 soil variables. Each feature set is fed into a separate cell of the LSTM model at layer 1.

For this study, all five aforementioned basic feature sets are utilized. Additionally, various combinations of numerical and imagery data are examined, including a combination of numerical data, combinations of one satellite imagery data with numerical data, combinations of two imagery datasets, combinations of two imagery datasets with numerical data, combination of three imagery datasets, and a combination of all possible features. This results in 17 distinct combinations of feature sets being input into the proposed model. The feature sets and their descriptions are detailed in Table 3.

## Evaluation of ResNet50-LSTM configurations across selected feature sets

This study focuses on both imagery and numerical data. Satellite variables–EVI, LAI, FPAR–are regarded as imagery data, whereas climate and soil variables are expressed as numerical values. Features from all imagery data are extracted using ResNet50 architecture. A total of 17 different combinations of input feature sets are created and fed into three proposed configurations of the LSTM model. Three configurations are presented to determine the effect of addition and connection of different layers on prediction outcomes. These configurations are briefly explained as follows:

### Model configuration 1: single LSTM layer without interconnected cells

In this configuration, there is a single LSTM layer, and each LSTM cell operates independently, characterized by minimal complexity. The model architecture is illustrated in Fig. 5A.

### Model configuration 2: single LSTM layer with interconnected cells

In this configuration, there is a single LSTM layer, and the LSTM cells within this layer are interconnected. The rest of the architecture is identical to that of model configuration 1. It allows the model to capture complex dependencies. The model architecture is illustrated in Fig. 5B.

### Model configuration 3: dual LSTM layers with interconnected cells

In this configuration, a deeper LSTM architecture is introduced having two LSTM layers. In the first LSTM layer, which receives the input, all cells are interconnected. It captures

**Table 3 Feature sets used in this study.**

| Serial no. | Feature set | Description of features |
| --- | --- | --- |
| 1 | FS01 | EVI |
| 2 | FS02 | FPAR |
| 3 | FS03 | LAI |
| 4 | FS04 | Climate variables |
| 5 | FS05 | Soil variables |
| 6 | FS06 | EVI + FPAR |
| 7 | FS07 | EVI + LAI |
| 8 | FS08 | LAI + FPAR |
| 9 | FS09 | Climate + Soil |
| 10 | FS10 | EVI + Climate + Soil |
| 11 | FS11 | FPAR + Climate + Soil |
| 12 | FS12 | LAI + Climate + Soil |
| 13 | FS13 | EVI + FPAR + LAI |
| 14 | FS14 | EVI + FPAR + Climate + Soil |
| 15 | FS15 | EVI + LAI + Climate + Soil |
| 16 | FS16 | LAI + FPAR + Climate + Soil |
| 17 | FS17 | EVI + LAI + FPAR + Climate + Soil |

initial temporal patterns and feeds them into the second layer, enhancing the model's capacity to learn hierarchical representations of sequential data. The rest of the architecture is identical to the model configuration two. The model architecture is illustrated in Fig. 5C.

Table 4 presents the validation results of all selected 17 feature sets for three proposed configurations of the model using $R^2$, RMSE, MAE, MBE, MRE, MAPE in percentage, and Willmott's index of agreement indicated by d. The bold entries show the best results among three model configurations and 17 feature sets.

The LSTM cells in the first layer are tailored according to feature sets. For instance, if a feature set contains only one set of features, then only one LSTM cell is employed in layer 1 to receive input. Consequently, for FS01–FS05, the model configurations 1 and 2 had the same effect and produced similar results because both configurations incorporate a single LSTM cell in layer 1, thereby diminishing the effect of interconnections between cells.

Figure 6 demonstrates the comparative analysis of all feature sets across the three configurations of the proposed model. It is evident from Fig. 6A that model configuration having dual LSTM layers yield higher $R^2$ values, except for FS04 and FS05, which consist of climate and soil variables respectively. These feature sets have a lower number of features, and do not produce satisfactory results. In model configuration 3, among all feature sets, FS14 demonstrates the highest $R^2$. Similarly, Willmott's index of agreement for third model configuration surpasses other configurations for all feature sets as illustrated in Fig. 6B. In particular, FS14 exhibits the highest value of d, signifying superior model performance of third model configuration for EVI, FPAR, climate, and soil variables.

**Table 4  Validation results of proposed configurations of ResNet50-LSTM model.**

| Feature set | Model configuration 1: single LSTM layer without interconnected cells | | | | | | | Model configuration 2: single LSTM layer with interconnected cell | | | | | | | Model configuration 3: dual LSTM layers with interconnected cells | | | | | | |
|---|---|---|---|---|---|---|---|---|---|---|---|---|---|---|---|---|---|---|---|---|---|
| | R² | RMSE | MAE | MBE | MRE | MAPE | d | R² | RMSE | MAE | MBE | MRE | MAPE | d | R² | RMSE | MAE | MBE | MRE | MAPE | d |
| FS01 | 0.7700 | 0.8236 | 0.6448 | 0.2873 | 0.0291 | 2.91% | 0.7417 | 0.7700 | 0.8236 | 0.6448 | 0.2873 | 0.0291 | 2.91% | 0.7417 | 0.9294 | 0.4686 | 0.3790 | -0.2028 | 0.0175 | 1.75% | 0.8617 |
| FS02 | 0.7164 | 1.0703 | 0.8090 | 0.4357 | 0.0354 | 3.54% | 0.7172 | 0.7164 | 1.0703 | 0.8090 | 0.4357 | 0.0354 | 3.54% | 0.7172 | 0.9305 | 0.4801 | 0.3564 | -0.0496 | 0.0164 | 1.64% | 0.8750 |
| FS03 | 0.6835 | 1.0447 | 0.7818 | 0.2364 | 0.0346 | 3.46% | 0.7015 | 0.6835 | 1.0447 | 0.7818 | 0.2364 | 0.0346 | 3.46% | 0.7015 | 0.9194 | 0.5033 | 0.3882 | 0.0193 | 0.0174 | 1.74% | 0.8620 |
| FS04 | 0.2102 | 1.6002 | 1.2152 | 0.2940 | 0.0528 | 5.28% | 0.4233 | 0.2102 | 1.6002 | 1.2152 | 0.2940 | 0.0528 | 5.28% | 0.4233 | 0.1770 | 1.5298 | 1.1355 | 0.0823 | 0.0511 | 5.11% | 0.4755 |
| FS05 | 0.0052 | 1.6702 | 1.3519 | -0.1595 | 0.0622 | 6.22% | 0.0969 | 0.0052 | 1.6702 | 1.3519 | -0.1595 | 0.0622 | 6.22% | 0.0969 | -0.0044 | 1.7610 | 1.4291 | 0.1964 | 0.0638 | 6.38% | 0.1366 |
| FS06 | 0.8723 | 0.6653 | 0.4976 | 0.0484 | 0.0226 | 2.26% | 0.8174 | 0.9146 | 0.5448 | 0.4457 | -0.1739 | 0.0204 | 2.04% | 0.8444 | 0.9832 | 0.2434 | 0.1871 | 0.1189 | 0.0084 | 0.84% | 0.9386 |
| FS07 | 0.8178 | 0.7853 | 0.6275 | -0.3069 | 0.0290 | 2.90% | 0.7543 | 0.9087 | 0.5535 | 0.4330 | 0.1456 | 0.0197 | 1.97% | 0.8416 | 0.9746 | 0.2707 | 0.2185 | -0.0442 | 0.0101 | 1.01% | 0.9171 |
| FS08 | 0.7854 | 0.7704 | 0.6370 | 0.4177 | 0.0281 | 2.81% | 0.7425 | 0.8878 | 0.5693 | 0.4385 | 0.0484 | 0.0200 | 2.00% | 0.8334 | 0.9579 | 0.3574 | 0.2803 | 0.1868 | 0.0124 | 1.24% | 0.8961 |
| FS09 | 0.2498 | 1.5720 | 1.2039 | 0.1352 | 0.0538 | 5.38% | 0.4593 | 0.2893 | 1.4193 | 1.1470 | -0.1708 | 0.0530 | 5.30% | 0.4602 | 0.3041 | 1.4099 | 1.1422 | 0.3876 | 0.0501 | 5.01% | 0.5379 |
| FS10 | 0.8038 | 0.8133 | 0.6382 | 0.2311 | 0.0285 | 2.85% | 0.7609 | 0.8458 | 0.6364 | 0.4838 | -0.2238 | 0.0221 | 2.21% | 0.8108 | 0.9777 | 0.2523 | 0.1772 | 0.0475 | 0.0080 | 0.80% | 0.9348 |
| FS11 | 0.8400 | 0.7434 | 0.5804 | 0.1130 | 0.0261 | 2.61% | 0.7941 | 0.8889 | 0.5495 | 0.4461 | -0.0569 | 0.0206 | 2.06% | 0.8218 | 0.9871 | 0.2175 | 0.1610 | 0.0201 | 0.0073 | 0.73% | 0.9491 |
| FS12 | 0.7248 | 0.8932 | 0.7217 | -0.3910 | 0.0336 | 3.36% | 0.7245 | 0.7754 | 0.8253 | 0.6372 | 0.4522 | 0.0283 | 2.83% | 0.7746 | 0.9502 | 0.3970 | 0.2758 | -0.2217 | 0.0126 | 1.26% | 0.9055 |
| FS13 | 0.8719 | 0.6441 | 0.5226 | 0.3051 | 0.0230 | 2.30% | 0.8081 | 0.9447 | 0.4108 | 0.3246 | -0.1720 | 0.0151 | 1.51% | 0.8788 | 0.9702 | 0.2890 | 0.2320 | -0.1751 | 0.0107 | 1.07% | 0.9153 |
| FS14 | 0.8948 | 0.5967 | 0.4768 | 0.1546 | 0.0214 | 2.14% | 0.8255 | 0.9450 | 0.4145 | 0.3340 | -0.1145 | 0.0153 | 1.53% | 0.8788 | **0.9903** | **0.1854** | **0.1384** | **0.0701** | **0.0062** | **0.62%** | **0.9536** |
| FS15 | 0.8795 | 0.6357 | 0.4658 | 0.1651 | 0.0205 | 2.05% | 0.8311 | 0.9556 | 0.4026 | 0.3278 | 0.1087 | 0.0148 | 1.48% | 0.8932 | 0.9705 | 0.2959 | 0.2346 | -0.1094 | 0.0106 | 1.06% | 0.9168 |
| FS16 | 0.8645 | 0.7205 | 0.5706 | 0.2369 | 0.0257 | 2.57% | 0.8001 | 0.9144 | 0.5217 | 0.4243 | -0.1951 | 0.0197 | 1.97% | 0.8375 | 0.9778 | 0.2594 | 0.2032 | -0.0680 | 0.0094 | 0.94% | 0.9258 |
| FS17 | 0.8509 | 0.7102 | 0.5891 | 0.5224 | 0.0260 | 2.60% | 0.7976 | 0.9315 | 0.4524 | 0.3493 | -0.0499 | 0.0159 | 1.59% | 0.8689 | 0.9737 | 0.3013 | 0.2274 | -0.1342 | 0.0105 | 1.05% | 0.9239 |

**Note:**
The bold entries show the best results among three model configurations and 17 feature sets.

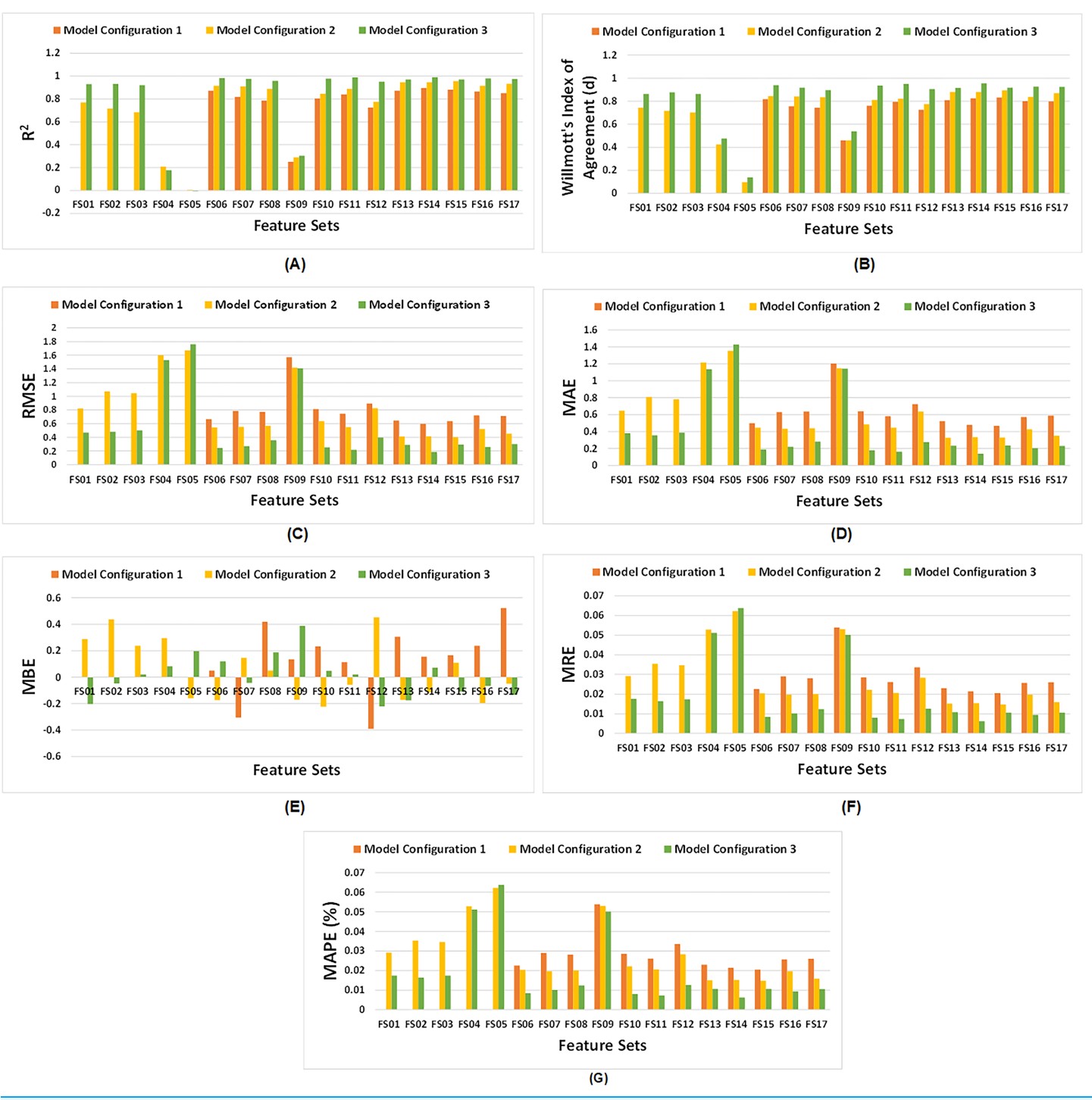

**Figure 6 Comparative analysis of all three configurations of proposed model with respect to feature sets.** (A) Comparative $R^2$, (B) comparative Willmott's index of agreement 'd', (C) comparative RMSE, (D) comparative MAE, (E) comparative MBE, (F) comparative MRE, and (G) comparative MAPE (%).

As depicted in Figs. 6C, 6D, 6F, and 6G, RMSE, MAE, MRE and MAPE of third model configuration decline as compared to other model configurations for all feature sets indicating satisfactory performance of model. But, for FS05, it did not perform well. However, for model configuration 3, FS14 exhibits the lowest values of RMSE, MAE, MRE and MAPE among all feature sets. Figure 6E illustrates that some feature sets result in underpredictions, while other lead to overpredictions.

However, it is observed that model configuration having dual LSTM layers with interconnected cells demonstrates the best performance among all configurations. Additionally, among the satellite variables (EVI, LAI, and FPAR), FPAR, represented as FS02, provides higher accuracy compared to the other satellite variables $i.e.$, $R^2$ = 0.9305. The combination of EVI and FPAR, represented as FS06, outperforms other combinations of two satellite variables $i.e.$, $R^2$ = 0.9832. For all satellite variables excluding environmental data, represented as FS13, results drop to $R^2$ = 0.9702. This exhibits that EVI and FPAR give favorable results when combined, particularly when integrated with climate and soil variables as demonstrated by FS14 with $R^2$ = 0.9903. The model operates effectively within 100 epochs and exhibits convergence. The convergence of model configuration 3 is illustrated in the form of a loss $vs.$ epochs plot for each feature set in Fig. 7.

Figure 8 presents scatter plots derived from reported values and predicted values produced by the third model configuration for all 17 feature sets. It is noticed that for FS04, FS05, and FS09, the model exhibited poor performance.

Summarizing, the model configuration having dual LSTM layers with interconnected cells outperforms other proposed configurations for all feature sets. Additionally, feature set FS14, which includes EVI, FPAR, climate, and soil variables, attains the best results among all feature sets. On the other hand, feature sets FS04 and FS05 performed the worst, likely due to their limited number of features.

## DISCUSSION

This study introduces a fusion of deep learning models, combining ResNet50 and LSTM, to predict rice yield. The model incorporates multi-modal data collected from various sources including three remotely sensed satellite variables: EVI, LAI, and FPAR, along with meteorological and soil parameters. The satellite images undergo feature extraction using the ResNet50 model as it is known for its effectiveness in feature extraction because of its residual learning mechanisms, generalization capabilities, and balanced complexity. Subsequently, these features, along with other environmental parameters are fed into the proposed LSTM model. LSTM is employed as a predictive model because it is a powerful sequence modeling technique. This study presents three configurations of the ResNet50-LSTM model. The architecture of each configuration slightly differs from each other. The ResNet50 excels at capturing hierarchical spatial features from satellite images, whereas LSTM can model temporal dependencies. The hybrid ResNet50-LSTM architecture leverage both spatial features (extracted by ResNet50) and temporal dynamics (modeled by LSTM), thus combining the strengths of both approaches.

The study concludes that the configuration with two LSTM layers, where all LSTM cells are connected to the previous cells of that layer, gives optimal results and converges

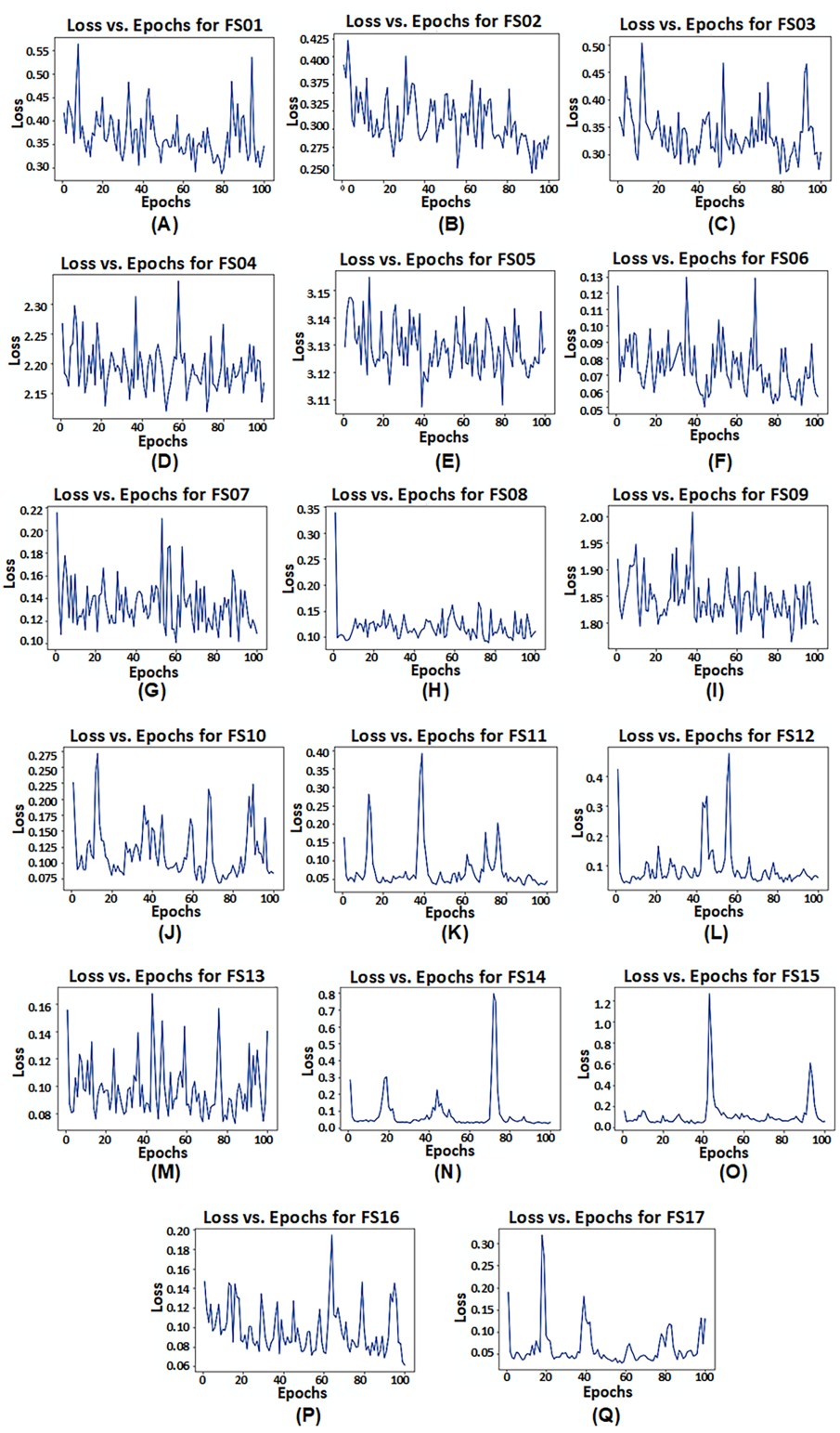

**Figure 7 Loss *vs.* epochs plot of model configuration having dual LSTM layers with interconnected cells for feature set FS01–FS17.**

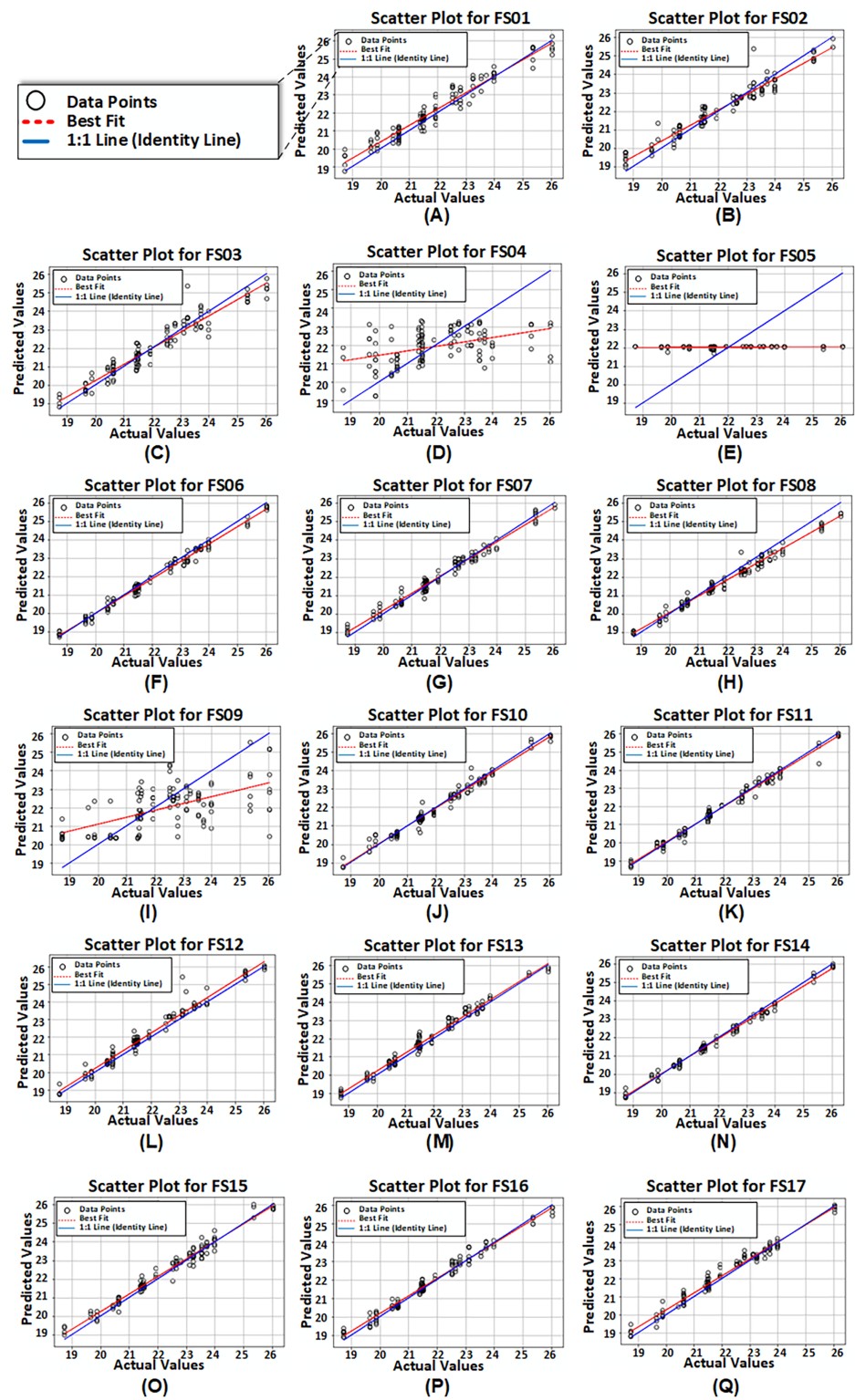

**Figure 8** (A-Q) Scatter plots of reported or actual values *vs.* predicted values computed from proposed model configuration 3 having dual LSTM layers with interconnected cells for feature set FS01–FS17.

quickly. The model was trained on a corei5 processor and 8-GB RAM. But, for feature sets FS13 (EVI, LAI, FPAR) and FS19 (all possible combinations), the model fails to operate on 8-GB RAM, due to their large number of trainable parameters and computational complexity. Consequently, for these combinations, we had to work with 16-GB RAM.

The study identifies that the FPAR satellite variable significantly contributes to yield prediction compared to other variables. Combining FPAR with EVI enhances results, while the combination of all satellite variables (EVI, LAI, FPAR) shows a slight drop. However, the best results are obtained when FPAR and EVI are combined with meteorological parameters and soil variables. The findings also highlight that meteorological and soil variables alone do not contain sufficient hidden features for effective yield prediction. However, when combined with satellite data especially EVI and FPAR, they contribute to achieving optimal results.

For rice yield prediction, numerous studies have employed ML models (*Siyal, Dempewolf & Becker-Reshef, 2015*; *Son et al., 2020*; *Guo et al., 2021*), neural networks (*Chu & Yu, 2020*), Attention based LSTM (AtLSTM) and transformer model (*Liu et al., 2022*), CNN, and LSTM model (*Cao et al., 2021*; *Zhou, Xu & Chen, 2023*). Some investigations, such as those by *Guo et al. (2021)* and *Fu, Tian & Zhan (2023)*, have delved into understanding the impact of phenological stages on rice yield. On the other hand, several studies have traditionally relied on explicit hand-crafted feature engineering in their ML approaches. Our research introduces a unique approach leveraging deep learning. We employ LSTM and transfer learning-based ResNet50 within an automated end-to-end prediction framework. Notably, no previous studies have used the combination of ResNet50 and LSTM for rice yield prediction. Furthermore, ResNet50 has not been utilized for feature extraction in this context. Additionally, our LSTM layer architecture is entirely different from the frameworks presented in prior studies, offering an innovative configuration that enhances the prediction accuracy and efficiency. A distinctive aspect of our study is the utilization of pre-calculated remotely sensed indices of satellite images, a departure from certain studies that undertake explicit operations for index extraction. Despite reported accuracies in existing studies using deep learning approaches reaching up to 93%, our proposed approach signifies a substantial leap forward, achieving an impressive 99% accuracy. In Table 5, a comprehensive summary of state-of-the-art studies is provided, with bold entries in the "Predictive Model Used" column emphasizing the superior performance of their respective methodologies. The last entries in the table present a concise overview of our proposed model, highlighting its advancements and remarkable accuracy.

Within the scope of this study, our satellite image preprocessing involves a series of critical steps aimed at optimizing the satellite input data for our predictive model. The process incorporates region geometry application, land cover masking, interpolation techniques, and mapping, culminating in the transformation of the raw data into 16-time steps for comprehensive analysis. Notably, our approach distinguishes itself from some studies (*Fernandez-Beltran et al., 2021*; *Son et al., 2020*) by opting not to employ cloud masking during preprocessing. Our model is deliberately trained with the inclusion of cloud effects. The remarkable performance metrics of our study, which were achieved

**Table 5 Comparison of our proposed approach with state-of-the-art studies.** Entries in bold indicate the model that performs best. In the last column, the results of the best respective model are given.

| Studies | Targeted region | Time coverage | Input parameters | Pre-processing techniques | Predictive model used | Best results |
|---|---|---|---|---|---|---|
| *Zhou, Xu & Chen (2023)* | Hubei province of China | 2000–2019 | Yield and boundary data, satellite data (EVI, SAVI, GPP), weather data | Masking, Conversion of remote sensing images to normalized histogram, use dummy variable to represent spatial heterogeneity | CNN, Convolutional LSTM, **CNN-LSTM** | R = 0.934 RMSE = 89878 MAE = 52802 |
| *Fu, Tian & Zhan (2023)* | Maha Sarakham, north-eastern Thailand | – | Optical images (MODIS), SAR images (Sentinel-1), ground-truth | Extraction of NDVI and EVI, Smoothing, making logistic curves, and then derived variables for regression model | RF regression model | $R^2 = 0.95$ RMSE = 0.06 ton/ha |
| *Liu et al. (2022)* | Northwest India | 2001–2016 | Yield statistics, climate indicators, satellite data (NDVI, EVI, NIRv, SIF) | Mapping of land cover type | AtLSTM, **Informer** model | $R^2 = 0.81$ RMSE = 0.41 (t/ha) |
| *Cao et al. (2021)* | China | 2001–2015 | Satellite variables (EVI, SIF), soil properties, weather features | Pearson correlation analysis (PCA) for extracting and combining features | LASSO, ML-RF, **DL-LSTM** | $R^2 = 0.87$ RMSE = 298.11 (kg/ha) |
| *Fernandez-Beltran et al. (2021)* | Nepal | 2006–2014 | Sentinel-2, Climate data, soil data, ground truth | Apply additional bands *i.e.*, NDVI and cloud masks to the sentinel-2 images | 3D CNN | RMSE = 89.03 (kg/ha) |
| *Guo et al. (2021)* | South China | 1981–2010 | Rice phenology, climate data, yield data | Partial correlation analyses to remove compound effects, made variable combinations, selection of combinations to get impact of preseason | Back propagation, **SVM**, RF | $R^2 = 0.33$ RMSE = 737 |
| *Chu & Yu (2020)* | China | 2015–2017 | Yield data, area data, meteorology data | Missing value interpolation, data normalization, hot code assignment to regions on basis of area data, descriptive statistics of data | BBI-Model | RMSE = 0.0057 MAE = 0.0044 MBE = −0.0062 MRE = 0.5784 |
| *Son et al. (2020)* | Taiwan | 2000–2018 | Yield statistics, satellite data (NDVI) | Masking using blue band of satellite images, linear interpolation | RF, **SVM** | RMSE = 8.7% MAE = 5.6 Willmott's d = 0.95 |
| *Siyal, Dempewolf & Becker-Reshef (2015)* | Larkana, Pakistan | 2006–2013 | Conventional crop reporting data, satellite images (NDVI, RVI) | Classification of rice/non-rice areas, calculation of missing data, and vegetation indices, conversion of Landsat digital numbers to top-of-atmosphere reflectance | Bagged seven decision trees (majority vote) | R2 = 0.925 RMSE = 80726 t MBE = −85016 t |
| **Our proposed approach** | **Gujranwala, Pakistan** | **2000–2021** | **Yield statistics, satellite variables (EVI, FPAR, LAI), meteorological and soil parameters** | **Filtering, interpolation, region geometry, land cover masking, unification of data to 16 time-steps** | ResNet50-LSTM | **$R^2 = 0.9903$ RMSE = 0.1854 MAE = 0.1384 MBE = 0.0701 MRE = 0.0062 MAPE = 0.62% Willmott's d = 0.9536** |

without the explicit cloud masking technique, reflects the outcome of this distinctive approach, demonstrating an $R^2$ of 0.9903, RMSE of 0.1854, and Willmott's index of agreement (d) reaching 0.9536. This distinctive aspect of our approach highlights the resilience of our model, by demonstrating its capability to effectively handle cloud-affected satellite imagery.

The EVI satellite index MOD13A2 has been employed by certain studies in the literature (*Cao et al., 2021*; *Zhou, Xu & Chen, 2023*; *Liu et al., 2022*), obtaining $R^2$ values up

to 0.87. Moreover, previous studies have not employed FPAR for rice yield prediction, despite its demonstrated effectiveness in predicting autumn crop yield with an $R^2$ of 0.973 (*Dang et al., 2021*). Similarly, LAI has been proven to be effective in predicting wheat yield, incorporating vegetation temperature condition index (VTCI) and phenological stages, which showcases optimal results (*Tian et al., 2021*). Our findings emphasize the significant impact of FPAR on rice yield prediction, revealing its effectiveness over LAI.

Furthermore, a hybrid model combining ResNet and LSTM has been utilized for the evaluation of growth status under drought and heat stress, reaching around 97% accuracy (*Xing et al., 2023*). This research asserts an entirely unique ResNet50-LSTM layer architecture achieving 99.03% accurate results for rice yield prediction and strengthen the study.

These findings highlight the significance of our research by introducing a scalable and cost-effective methodology for predicting rice yields across large geographic regions in a timely manner. The approach is cost-effective as it utilizes remotely sensed, readily available real-time data, thereby reducing the requirement for extensive financial resources, and excessive operational costs. By utilizing publicly available multi-source data, our approach enhances the accessibility and applicability, ensuring practicality in diverse agricultural landscapes. The adaptability and effectiveness of our model hold promise for influencing agricultural practices on a global scale, promoting more informed decision-making and resource management strategies.

## CONCLUSION

This study introduces and evaluates three configurations of a deep learning-based predictive model that seamlessly integrates ResNet50 and LSTM models. The model is trained using multi-modal data from diverse sources, encompassing satellite data such as EVI, FPAR, and LAI indices, as well as climate and soil data. Our findings imply that the LSTM model configuration featuring two LSTM layers with interconnected cells emerges as the most efficient among the proposed configurations, delivering optimal results with rapid convergence during training. Exploring various feature combinations demonstrates that the amalgamation of FPAR, EVI, climate, and soil variables achieves the highest accuracy rate of 99% with minimal error rates. FPAR emerges as a key influencer in rice yield predictions, highlighting its significant impact on the model's performance. This research presents a cost-effective and scalable approach leveraging readily available real-time remotely sensed data for accurately predicting rice yields across vast regions, with potential for global utilization in crop yield estimation. These outcomes hold practical applications in crop management and planning, precision agriculture, climate change adaption, and risk assessment and management. Notably, our focus on MODIS satellite imagery lays the groundwork for future undertakings. Because of the high spatial resolution of Sentinel or Landsat products, we plan to extend this work by encompassing these products, thereby broadening the scope of our predictive model.

# ACKNOWLEDGEMENTS

The authors express sincere gratitude to Mr. Shahid Abbas, Chief Meteorologist of the Pakistan meteorological department (PMD), for his invaluable assistance and collaboration. His profound knowledge of the subject, and readiness to share his expertise and experiences has greatly contributed to the success of this research.

### Funding

The authors received no funding for this work.

### Competing Interests

The authors declare that they have no competing interests.

### Author Contributions

- Aqsa Aslam performed the experiments, analyzed the data, performed the computation work, prepared figures and/or tables, authored or reviewed drafts of the article, and approved the final draft.
- Saima Farhan conceived and designed the experiments, performed the experiments, analyzed the data, prepared figures and/or tables, authored or reviewed drafts of the article, and approved the final draft.

### Data Availability

The code is available at GitHub and Zenodo:

- https://github.com/AqsaAsla/Enhancing-Rice-Yield-Prediction-code.git
- AqsaAsla. (2024). AqsaAsla/Enhancing-Rice-Yield-Prediction-code: Initial Release (0.1.0). Zenodo. https://doi.org/10.5281/zenodo.12244347

The raw data is available at the following sites:

- Yield labels: Rice Estimates, Crop Reporting Service, Government of the Punjab, https://crs-agripunjab.punjab.gov.pk/node/165#overlay-context=reports

- The MOD13A2 v061, EVI: Didan, K. 2021. MODIS/Terra Vegetation Indices 16-Day L3 Global 1km SIN Grid V061 [Data set]. NASA EOSDIS Land Processes Distributed Active Archive Center. https://doi.org/10.5067/MODIS/MOD13A2.061

- The MOD15A2H v061, LAI, FPAR: Myneni, R., Knyazikhin, Y., Park, T. 2021. MODIS/Terra Leaf Area Index/FPAR 8-Day L4 Global 500m SIN Grid V061 [Data set]. NASA EOSDIS Land Processes Distributed Active Archive Center. https://doi.org/10.5067/MODIS/MOD15A2H.061

- Climate and Soil data: The POWER Project, NASA, https://power.larc.nasa.gov/data-access-viewer/

- Auxiliary data, MCD12Q1 v006, Land cover type product: Friedl, M., Sulla-Menashe, D. 2019. MCD12Q1 MODIS/Terra+Aqua Land Cover Type Yearly L3 Global 500m SIN Grid V006 [Data set]. NASA EOSDIS Land Processes Distributed Active Archive Center. https://doi.org/10.5067/MODIS/MCD12Q1.006

- Shape files: Pakistan-Subnational Administrative Boundaries, The Humanitarian Data Exchange, https://data.humdata.org/dataset/cod-ab-pak?

The yield data collected from Crop Reporting Service, Processed climate and soil data, and Google Earth Engine Java Scripts for the collection and processing of satellite imagery available at GitHub and Zenodo:

- https://github.com/AqsaAsla/Enhancing-Rice-Yield-Prediction-Data.git
- AqsaAsla. (2024). AqsaAsla/Enhancing-Rice-Yield-Prediction-Data: Initial Release (0.1.0). Zenodo. https://zenodo.org/records/12301300.

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
