# Peer review of "Enhancing rice yield prediction: a deep fusion model integrating ResNet50-LSTM with multi source data"

_PeerJ Computer Science, doi:10.7717/peerj-cs.2219_

## Round 0.1 · original submission · Major Revisions

· Academic Editor

Major Revisions

In addition to responding to all the comments, be sure to highlight the motivation and problems statement for this work.

Reviewer 1 ·

Basic reporting

The introduction section needs to be more focused and concise. The background information provided is extensive, but it could be streamlined to highlight the research gap and problem statement more effectively.

Experimental design

Provide a more detailed explanation and justification for the choice of the ResNet50-LSTM model architecture and the three configurations proposed. How do these configurations differ from existing approaches, and what are their potential advantages?
The dataset preprocessing steps are well-described, but it would be beneficial to include a table or diagram summarizing the various data sources, resolution, and temporal coverage for better clarity.
Clarify the rationale behind choosing the specific evaluation metrics used in the study. Additionally, provide a brief explanation of each metric's significance and interpretation for readers who may be unfamiliar with them.

Validity of the findings

The results section presents a comprehensive analysis of the different feature combinations and model configurations. However, it could benefit from a more organized and structured presentation, perhaps using subsections or clear headings to guide the reader through the findings.
Discuss the potential limitations or challenges associated with the proposed approach, such as computational complexity, scalability, or data availability constraints, and suggest possible solutions or future research directions.
Provide a more detailed comparison with existing state-of-the-art methods in the discussion section, highlighting the strengths and weaknesses of your approach relative to other techniques.
Expand the conclusion section to emphasize the key contributions and implications of your study, as well as potential applications or extensions of the proposed methodology.

Additional comments

Minor Comments:

Ensure consistent formatting and adherence to the journal's guidelines throughout the manuscript, including figures, tables, and references.
Proofread the manuscript carefully for any grammatical, spelling, or typographical errors that may have been overlooked.
Consider providing a brief description or summary of the study area's characteristics, such as climate, topography, or soil conditions, as these factors can influence rice yield.
Clarify the time period or crop season covered by the yield data used in the study (e.g., annual, seasonal, or specific months).
Explain any assumptions or considerations made regarding the handling or treatment of outliers or missing data in the dataset, if applicable.

Reviewer 2 ·

Basic reporting

The study presents a good integration of ResNet50 and LSTM for rice yield prediction, which is commendable. The potential impact of this research on practical applications in agriculture is significant. The paper makes a valuable contribution to the field of agricultural predictive modeling. Addressing the aforementioned comments will enhance the clarity, reproducibility, and impact of the work. However, there are some suggestions:
1. Write your objectives, contributions in proper way in introduction
2. You have not highlighted the problems in existing studies in introduction part.
3. Word Novel is not suitable
4. Abstract is too long, concise to upto 250 words and shift rest of the work to introduction
5. Approach is Cost effective in terms of what ?
6. Revise your results section such that divide it into different subsections as per your experiments to enhance the clarity of the section.
7. Methodology section is not providing an insightful elaboration of the proposed framework.
8. The manuscript maintains a good standard of English but could benefit from simplification of complex sentences for enhanced readability.
9. Minor grammatical errors and awkward phrasing need correction, e.g., "affected production" should be "affected the production."
10. Ensure consistency in terminology throughout the paper, such as consistently referring to "ResNet50" as "residual neural network (ResNet50)" or "ResNet50."
11. The methodology section requires additional clarification, especially in the data preprocessing steps (lines 28-32) to allow for replication.
12. Provide a rationale for the specific configurations and architectures of the LSTM models, and consider comparing with alternative configurations to strengthen the study.
13. Discuss the criteria used for selecting feature combinations and their impact on model performance in more detail.

Experimental design

14. The validation technique (10-fold cross-validation) is appropriate but needs more detail on data splitting and potential data leakage issues.
15. While the performance metrics used are comprehensive, a detailed discussion on why these metrics were chosen and their comparison with metrics used in similar studies would add value.
16. Include a detailed comparison with baseline or state-of-the-art models on the same dataset to validate the claims.
17. Discuss the quality and resolution of data from multiple sources (MODIS, NASA POWER), any limitations, and how they were addressed.

Validity of the findings

no comment

---

## Round 0.2 · accepted · Accept

· Academic Editor

Accept

The paper has been revised according to the reviewer’s comments.

Reviewer 1 ·

Basic reporting

The authors have addressed all the comments.

Experimental design

The authors have addressed all the comments.

Validity of the findings

The authors have addressed all the comments.

Additional comments

The authors have addressed all the comments.

Reviewer 2 ·

Basic reporting

The authors have successfully addressed my comments.

Experimental design

no comment

Validity of the findings

no comment